# Investigating the Mechanical Properties of ZrO_2_-Impregnated PMMA Nanocomposite for Denture-Based Applications

**DOI:** 10.3390/ma12081344

**Published:** 2019-04-25

**Authors:** Saleh Zidan, Nikolaos Silikas, Abdulaziz Alhotan, Julfikar Haider, Julian Yates

**Affiliations:** 1Dentistry, School of Medical Sciences, University of Manchester, Manchester M13 9PL, UK; Nikolaos.Silikas@manchester.ac.uk (N.S.); abdulaziz.alhotan@postgrad.manchester.ac.uk (A.A.); Julian.yates@manchester.ac.uk (J.Y.); 2School of Engineering, Manchester Metropolitan University, Manchester M1 5GD UK; j.haider@mmu.ac.uk

**Keywords:** PMMA, zirconia (ZrO_2_), nanocomposite, denture base, flexural strength, impact strength, fracture toughness, hardness

## Abstract

Acrylic resin PMMA (poly-methyl methacrylate) is used in the manufacture of denture bases but its mechanical properties can be deficient in this role. This study investigated the mechanical properties (flexural strength, fracture toughness, impact strength, and hardness) and fracture behavior of a commercial, high impact (HI), heat-cured denture base acrylic resin impregnated with different concentrations of yttria-stabilized zirconia (ZrO_2_) nanoparticles. Six groups were prepared having different wt% concentrations of ZrO_2_ nanoparticles: 0% (control), 1.5%, 3%, 5%, 7%, and 10%, respectively. Flexural strength and flexural modulus were measured using a three-point bending test and surface hardness was evaluated using the Vickers hardness test. Fracture toughness and impact strength were evaluated using a single edge bending test and Charpy impact instrument. The fractured surfaces of impact test specimens were also observed using a scanning electron microscope (SEM). Statistical analyses were conducted on the data obtained from the experiments. The mean flexural strength of ZrO_2_/PMMA nanocomposites (84 ± 6 MPa) at 3 wt% zirconia was significantly greater than that of the control group (72 ± 9 MPa) (*p* < 0.05). The mean flexural modulus was also significantly improved with different concentrations of zirconia when compared to the control group, with 5 wt% zirconia demonstrating the largest (23%) improvement. The mean fracture toughness increased in the group containing 5 wt% zirconia compared to the control group, but it was not significant. However, the median impact strength for all groups containing zirconia generally decreased when compared to the control group. Vickers hardness (HV) values significantly increased with an increase in ZrO_2_ content, with the highest values obtained at 10 wt%, at 0 day (22.9 HV_0.05_) in dry conditions when compared to the values obtained after immersing the specimens for seven days (18.4 HV_0.05_) and 45 days (16.3 HV_0.05_) in distilled water. Incorporation of ZrO_2_ nanoparticles into high impact PMMA resin significantly improved flexural strength, flexural modulus, fracture toughness and surface hardness, with an optimum concentration of 3–5 wt% zirconia. However, the impact strength of the nanocomposites decreased, apart from the 5 wt% zirconia group.

## 1. Introduction

In practical applications, denture base materials experiences different types of stresses, such as compressive, tensile and shear, which can lead to premature failure. Intra-orally, repeated mastication over a period of time can lead to denture base fatigue failure. Extra-orally, denture bases can also experience high impact forces when dropped by accident [1,2]. Impact fractures occur extra-orally as a result of inadvertent denture damage [1,3]. The incidence of denture fracture is relatively high: 68% of dentures fail within three years of fabrication and the incidence in partial denture is greater than that of complete dentures [4,5]. Studies have also reported that 33% of the repairs in dental laboratories are as a result of de-bonded teeth, and 29% percent of fractures occur in the midline of the denture base, being seen more frequently in the upper than in the lower prosthesis [6,7]. The remaining 38% of fractures are caused by other types of failure [6,7].

High impact (HI) denture base resins are widely used in prosthetic dentistry. These materials are provided in either powder or liquid forms and are processed in the same manner as other heat-cured, poly-methyl methacrylate (PMMA) resins. HI resins are reinforced with butadiene-styrene rubber, with the rubber particles grafted to the poly-methyl methacrylate so that the particles are covalently bonded into the polymerized acrylic matrix in order to better absorb mechanical loads [4,8,9,10]. Incorporation of butadiene-styrene rubber into PMMA resins improves impact strength and dimensional stability [8,11,12]. However, such reinforcement can result in the reduction of mechanical properties, including flexural strength, fatigue strength and stiffness [8,11,13].

Many attempts have been made to improve the strength of denture base resins, including the addition of metal wires and plates made of either Co-Cr alloy or stainless steel. However, these materials present limitations contrary to the standard requirements, including poor adhesion between the acrylic resin and reinforcing metal. This separation can result in a reduction in overall mechanical strength within the prosthesis, as well as poor aesthetics. Additionally, metal-reinforced denture bases can become noticeably heavier [13,14]. Other attempts to improve denture base mechanical properties include fibre reinforcement to enhance fracture toughness, flexural and impact strength, and fatigue properties [13,15]. Different fibre types, such as ultra-high modulus polyethylene fibre (UHMPE), aramid fibre, nylon fibre, carbon fibre and glass fibre, have all been investigated [13,15,16]. UHMPE fibre does not demonstrate good adhesion to PMMA, and therefore, no significant increase in flexural properties has been demonstrated [17]. Carbon and aramid fibres are not practical materials because of difficulties in polishing the final prostheses, and resultant poor aesthetics [18]. However, nylon reinforcement enhances fracture resistance and structural elasticity of acrylic resins [15]. A study undertaken by Vallittu et al., on the flexural and transverse strength of heat-cured PMMA denture bases reinforced with a high concentration of continuous glass fibre demonstrated an improvement in these properties [19]. Additionally, silane coupling agents have been added to enhance adhesion between the polymer resin and glass fibres to improve mechanical strength, resulting in enhanced flexural and fatigue strength [19,20]. However, fibre orientation in the resin matrix is technically difficult to control and a random distribution could result in defects within the finished product [21].

In recent years, several investigations have focused on improving the mechanical properties of PMMA acrylic resins by adding nanomaterials, such as bio-ceramic nanoparticles, due to their special characteristics [22]. Zirconia (ZrO_2_) is a bio-ceramic material that has been widely used for various dental applications, such as crowns and bridges, implant fixture “screws” and abutments, and orthodontic brackets [23]. Zirconia has a high flexural strength (900 to 1200 MPa), hardness (1200 HV), and fracture toughness (9–10 MPa m^1/2^) [24]. Furthermore, zirconia shows excellent biocompatibility compared to other ceramic materials, such as alumina [22,24]. A number of studies found that reinforcement of conventional, heat-cured denture base resins with zirconia nanoparticles significantly improved mechanical properties such as flexural and impact strength, as well as surface hardness [22,25]. However, no systematic study on the effect of zirconia addition in the high impact (HI) heat-cured PMMA denture base material has been reported in the literature. Therefore, research is needed to identify an optimum amount of zirconia suitable for improving performance and life of HI PMMA denture bases. 

The purpose of this study is to evaluate the effects of zirconia nanoparticle addition at low concentrations (up to 10%) to a commercially available, high-impact, PMMA denture base resin on selected mechanical properties such as flexural strength, impact strength, fracture toughness, hardness and fracture behaviour. 

## 2. Material and Methods

### 2.1. Materials 

A commercially available, Metrocryl HI denture base powder, (PMMA, poly-methyl methacrylate) and Metrocryl HI (X-linked) denture base liquid (MMA, methyl methacrylate) (Metrodent Limited, Huddersfield, UK) were selected as the denture base material. Yttria-stabilized zirconia (ZrO_2_) nanoparticles (94% purity; Sky Spring Nano materials, Inc., Houston, TX, USA) were chosen as the inorganic filler agent for fabricating the nanocomposite denture base specimens.

### 2.2. Specimen Preparation

#### 2.2.1. Silane Functionalization of Zirconia Nanoparticle Surfaces 

Fifteen grams of zirconia nanoparticles and 70 mL of toluene solvent were deposited into a plastic container, which was then placed in a speed mixer (DAC 150.1 FVZK, High Wycombe, UK), and mixed at 1500 rpm for 20 min. Following the initial mixing, 7 wt% silane coupling agent (3-trimethoxysilyl propyl methacrylate; product no. 440159, Sigma Aldrich, Gillingham, UK) was added slowly over a period of 20 s. The mixture was then placed in the speed mixer at 1500 rpm for 10 min and divided equally into two tubes and spun in a centrifuge at 23 °C at 4000 rpm for 20 min. The supernatant (separated toluene) was removed, and the remaining silanized nanoparticles were transferred into a personal solvent evaporator (EZ-2 Elite, Genevac Ltd., SP Scientific Company, Ipswich, UK) for 3 h of drying at 60 °C.

#### 2.2.2. Selection of Appropriate Percentages of Zirconia Nanoparticles

To determine the most appropriate weight percentages of zirconia nanoparticles for the current study, preliminary investigations were undertaken using 1.5 wt%, 10 wt% and 15 wt% mixtures. Based on these results and knowledge from relevant literature, a decision was made to utilize the following weight percentages of silanized zirconia nanoparticles in the denture base formulation: 0.0% (control), 1.5 wt%, 3.0 wt%, 5.0 wt%, 7.0 wt%, and 10.0 wt%. The composition details of the specimen groups used in this study are described in Table 1 (all used an acrylic resin powder:monomer ratio of 21 g:10 mL, in accordance with manufacturer’s instructions). 

#### 2.2.3. Mixing of Zirconia with PMMA 

The silane-treated zirconia and acrylic resin powders were weighed according to Table 1 using an electronic balance (Ohaus Analytical with accuracy up to 3 decimal points). The zirconia powder was added to the acrylic resin monomer and mixed by hand using a stainless-steel spatula to make sure all the powder was uniformly distributed within the resin monomer. The HI acrylic resin powder was then added to the solution, and mixing continued until a consistent mixture was obtained, according to the manufacturer’s instruction. The mixing continued for approximately 20 min until the mixture reached a dough-like stage, which was suitable for handling. When the mixture reached a consistent dough-like stage (working stage), it was packed into a mould by hand. The moulds were made from aluminium alloy, which contained five cavities with a dimension of 65 mm (l) × 10 mm (w) × 2.50 mm (d) for producing flexural strength and hardness test samples. However, the cavity dimensions for the impact test was as follows: 80 mm (l) × 10 mm (w) × 4 mm (d) and fracture toughness was 40 mm (l) × 8 mm (w) × 4 mm (d). Before pouring the mixture into the mould, sodium alginate as a separating medium (John Winter, Germany) was applied to the surfaces of the mould for easy removal of the specimens. The mould was then closed and placed in a hydraulic press (Sirio P400/13045) under a pressure of 15 MPa in the first cycle, and then the pressure was released. Excess mixture was removed from the mould periphery, which was then re-pressed at room temperature for 15 min under the same pressure. The mould was then immersed in a temperature-controlled curing water bath for 6 h to allow polymerization. The curing cycle involved increasing the temperature to 60 °C over 1 h and maintained this temperature for 3 h. After this time, the temperature was increased to 95 °C over an additional 2 h to complete the heat polymerization cycle. The mould was removed from the curing bath and cooled slowly for 30 min at room temperature. The mould was then opened and the specimens were removed. The specimens were then trimmed using a tungsten carbide bur, ground with an emery paper and polished with pumice powder in a polishing machine (Tavom, Wigan, UK) in accordance with British International Standard Organization (BS EN ISO 20795-1:2008) and British Standard Specification for Denture Base Polymers (BS 2487: 1989 ISO 1567; 1988) [26,27].

### 2.3. Mechanical Characterization of the Nanocomposite

#### 2.3.1. Flexural Strength Test

Flexural strength of the nanocomposite specimens was evaluated using a 3-point bend test in a universal testing machine (Zwick/Roell Z020 Leominster, UK) in accordance with British International Standard for Denture Base Polymers (2487: 1989) [27]. The dimensions of the specimens were 65 mm length × 10 ± 0.01 mm width × 2.50 ± 0.01 mm thickness. All specimens were stored in distilled water at a temperature of 37 ± 1 °C for 50 ± 2 h in an incubator before testing. The specimens were then removed from the distilled water and placed on a support jig. The loading plunger (diameter 7.0 mm) was fixed at the center of the specimen midway between two supports, which were parallel and separated by 50 ± 0.1 mm, and the diameter of the load supports were 3.20 mm. A 500 N load cell was used to record force and the load was applied using a cross-head speed of 5 mm/min. The maximum force (F) was recorded in newtons, and flexural strength was calculated in MPa for all specimens using the following equation [28]:(1)$$\sigma =\frac{3\mathrm{Fl}}{2{\mathrm{bh}}^{2}}$$
where F is the maximum force applied in N, l is the distance between the supports in mm, b is the width of the specimen in mm, and h is the height of the specimen in mm. The flexural modulus was determined as the slope of the linear portion of the stress/strain curve for each test run.

#### 2.3.2. Fracture Toughness Test 

Fracture toughness tests were conducted using a single edge span notch bending test on the Zwick universal testing machine in accordance with the British International Standard Organization (BS EN ISO 20795-1:2008) [26,29]. The dimensions of the specimens were 40 mm (l) × 8 mm (w) × 4 mm (h), and a notch was created in the middle of the specimens with a diamond blade and a saw to a depth of 3.0 ± 0.2 mm along a marked centre line. All specimens were then stored in distilled water and placed in an incubator at 37 ± 1 °C for 168 ± 2 h before testing. The specimens were removed from the water, dried by a towel and placed edgewise on the supports of the testing rig. The notch of the specimen was placed directly opposite to the load plunger (diameter 7 mm) and in the middle of the span between the two supports (32.0 ± 0.1 mm). The load cell was 500 N, and the cross-head speed was 1.0 mm/min. Fracture toughness was determined by increasing the force from zero to a maximum value in order to propagate a crack from the opposite side of the specimen to the impact point. The maximum force (P) in newtons to fracture was recorded in order to calculate the fracture toughness (K_IC_) in MPa m^1/2^ according to Equation (2) [29]:(2)KIC=3PL2BW32×Y
where W is the height of the specimen in mm, B is the width of the specimen in mm, L is the distance between the supports in mm, and Y is a geometrical function calculated by Equation (3).
(3)$$Y=1.93\;\times {\left(\frac{a}{w}\right)}^{1/2}-\;3.07\;\times {\left(\frac{a}{w}\right)}^{\frac{3}{2}}+14.53\;\times {\left(\frac{a}{w}\right)}^{\frac{5}{2}}-25.11\;\times {\left(\frac{a}{w}\right)}^{\frac{7}{2}}+25.80\;\times {\left(\frac{a}{w}\right)}^{\frac{9}{2}}$$
where a is the depth of the notch.

#### 2.3.3. Impact Test 

The Charpy V-notch impact test (kJ/m^2^) utilized a universal pendulum impact testing machine (Zwick/Roell Z020 Leominster). Specimen dimensions were 80 mm (l) × 10 ± 0.01 mm (w) × 4 ± 0.01 mm (h), in accordance with the European International Standard Organization (EN ISO 179-1:2000) [30]. The specimens were notched in the middle to a depth of 2.0 ± 0.2 mm, a notch angle of 45 ° and a notch radius of 1.0 ± 0.05 mm and were then stored in distilled water at 37 ± 1 °C for 168 ± 2 h in an incubator before testing. The specimens were then removed from the water and dried with a towel. Each specimen was placed in the machine and were supported horizontally at its ends (40 ± 0.2 mm), and the centre of the specimen (the un-notched surface) was hit by a free-swinging pendulum that was released from a fixed height. The pendulum load cell was 0.5 J and directly faced the centre of the specimen, as shown in Figure 1. When the test was started, the pendulum was released to strike the specimen, and the impact energy absorbed was recorded in joules (J). The Charpy impact strength ($a_{\mathrm{iN}}$) (kJ/m^2^) was calculated using Equation (4) [10,30]: (4)$$a_{\mathrm{iN}}=\frac{E_{c}}{{h*b}_{N\;}}\;\times \;10^{3}$$
where E_c_ is the breaking energy in joules absorbed by breaking, h is the thickness in mm, and b_N_ is the remaining width in mm after notching.

#### 2.3.4. Hardness Test 

The Vickers hardness (HV_0.05_) of the specimens was measured using a micro-hardness testing machine (FM-700, Future Tech Corp, Tokyo, Japan). Specimens were 65 mm length × 10 mm width × 2.50 mm thickness, and the test load was fixed at 50 g for 30 s. The Vickers hardness was calculated by measuring the diagonals of the pyramid-shaped indentation impressed on the specimen. A total of three indentations were taken at different points in each specimen one side, and then a mean value was calculated. The mean hardness values for all the specimens were determined demonstrative of the materials in the dry condition at day 0. The specimens were then stored individually in 37 ± 1 °C distilled water for 7 d ± 2 h, and were then re-immersed for a total of 45 d ± 2 h. From the raw data, the mean hardness values for each sample group were calculated [31,32].

### 2.4. Scanning Electron Microscopy (SEM) Examination

The size and shape distribution of the PMMA powder and zirconia nanoparticles was analysed using a scanning electron microscope (SEM) (Carl Zeiss Ltd, 40 VP, Smart SEM, Cambridge, UK). The fractured surface was also studied to identify failure mechanism. Specimens were mounted onto aluminium stubs and sputter-coated with gold after which SEM visualization was performed using a secondary electron detector at an acceleration voltage of 2.0 kV. 

### 2.5. Statistical Analyses 

Flexural strength, modulus, impact strength, fracture toughness and Vickers hardness data were analysed using a statistical software (SPSS statistics version 23, IBM, New York, NY, USA). Non-significant Shapiro–Wilk and Levene tests showed that the data of flexural and fracture toughness were normally distributed and there was homogeneity of variance. The flexural and fracture toughness data were analysed using a one-way analysis of variance (ANOVA) with the Tukey honestly significant difference post-hoc test at a pre-set alpha of 0.05. Impact strength and hardness data demonstrated nonparametric distributions as evidenced by significant Shapiro–Wilk test results for two groups, and therefore the Kruskal–Wallis test was used to analyse the results as well as to compare the differences among the test groups at a pre-set alpha of 0.05. In addition, the Friedman’s two-way analysis test was applied to identify any significant difference between the three immersion time groups (*p* < 0.05).

## 3. Results 

### 3.1. Visual Analysis

SEM analysis revealed that the average particle size of the PMMA powder was approximately 50 µm with a range from 10 µm to 100 µm, as shown in Figure 2A. The rubber particles were also visible within the powder, with an average size of approximately 50 µm. The as-received, yttria-stabilized zirconia nanoparticles demonstrated an average size ranging between 30 nm and 60 nm for individual particles and 200 nm to 300 nm for clusters, as shown in Figure 2B.

### 3.2. Mechanical Tests

#### 3.2.1. Flexural Strength and Flexural Modulus

One-way analysis of variance (ANOVA) of flexural strength values presented in Table 2 show a significant difference (*p* < 0.05) for the specimen group containing 3 wt% zirconia. However, the mean values of flexural modulus showed a significant increase *(p* < 0.05) for all specimens, except that containing 7 wt% zirconia, which was not significantly different (*p* > 0.05) from the control group. The flexural strength data in the table demonstrates that an addition of zirconia nanoparticles to the HI PMMA gradually increased the strength up to 3 wt% and then gradually decreased for other compositions when compared to the control group (0 wt%). The highest value of flexural strength was recorded for the group containing 3 wt% zirconia (83.5 MPa) in comparison with the control group (72.4 MPa), representing a 15% increase in the flexural strength. However, a higher percentage of zirconia nanoparticles (7 wt% to 10 wt%) in the specimens reduced the strength, which was comparable to the control group. A similar behaviour was also found for the flexural modulus of the nanocomposites with increasing zirconia content (Table 2). However, a maximum value of the flexural modulus was reached at a zirconia content of 5 wt% (2419 MPa) when compared to the control group (1971 MPa), meaning an increase of 22.7%. Furthermore, even though at high zirconia content the modulus values decreased, they were still higher than those of the control group.

#### 3.2.2. Fracture Toughness and Impact Strength

The mean values of the fracture toughness (Table 2) of the nanocomposites decreased significantly compared to that of the control group at the zirconia concentrations of 7% and 10% (*p* < 0.05). Furthermore, after the initial decrease of fracture toughness at 1.5 wt% zirconia, the values slightly increased in the groups containing 3 wt% and 5 wt% zirconia, but they were not statistically significant increases (*p* > 0.05). Table 2 shows that the best fracture toughness could be achieved at 5 wt% zirconia.

The values of the impact strength for all nanocomposite groups were not statistically significant *(p* > 0.05), as shown in Table 2. The median impact strength gradually decreased with the increase in zirconia content, except in the group containing 5 wt% zirconia, which showed the best impact strength (only 10% reduction compared to the control group). However, all measured impact strength values for the nanocomposites were lower than that for the control group.

#### 3.2.3. Hardness

The median values of Vickers hardness in Table 3 show significant differences (*p* < 0.05) for the specimen groups containing 7 wt% and 10 wt% zirconia in both dry (0 day) and wet (7 days) conditions. From the graphical presentation of the hardness results (Figure 3), it is interesting to note that at lower zirconia contents (1.5–5.0%), the difference in hardness between dry and wet conditions was much lower than that at higher zirconia contents (7.0–10.0%). Furthermore, no significant difference was found between the hardness of the specimens stored in water for seven days and 45 days at all zirconia contents. This finding indicates that the hardness of the nanocomposites does not degrade over time in the wet condition at lower zirconia contents, particularly up to 3% zirconia.

### 3.3. Microstructural Characteristics

The fractured surface of pure PMMA specimens displayed a smooth surface in small areas and revealed a ductile type failure behaviour exhibiting irregular and rough surface as is shown in Figure 4A. The composite fractured surface showed signs of cracks and particle clustering with small voids (Figure 4B). Figure 4C presents more clear fracture features and shows that the distribution of the nanoparticles was not uniform. The image highlights particle clustering in several places and voids on the fractured surface.

## 4. Discussion

In this study, it was shown that combining zirconia nanoparticles to HI acrylic resin improved flexural strength and flexural modulus, which can lead to a reduction in different types of stresses encountered during the mastication process, including compressive, tensile and shear stresses [33]. However, the reinforced HI acrylic resin with lower concentration of zirconia (5%) did not show any significant difference from the control group on fracture toughness and impact strength. 

The inorganic reinforcing nano-fillers have a large surface area that provides high surface energy, and this produces nanoparticles with a strong tendency to aggregate. This characteristic may decrease the chemical interaction between the nanoparticles and the base PMMA [22]. In this study, to enhance the chemical adhesion between the ZrO_2_ nanoparticles and ZrO_2_-PMMA, the surface of the ZrO_2_ particles was treated with 7 wt% silane coupling agent (3-MPS) to create reactive functional groups. This could be responsible for improving the flexural properties of the nanocomposites at lower concentrations of zirconia nanoparticles. Moreover, the improvement in flexural strength and flexural modulus could be a result of the improved dispersion of the ZrO_2_ nanoparticles when mixing with the speed mixer machine during the preparation stage. This improvement would decrease the agglomeration tendency in the composites. Additionally, the large interfacial area of the nanoparticles contributes to more contact points between the ZrO_2_ and PMMA, thus enhancing mechanical interlocking and offers additional flexibility in the nanocomposites [34].

Only a few studies on the effect of adding ZrO_2_ nanoparticles in HI heat-cured denture base acrylic resin are available in the literature. In contrast, investigators have worked on improving the mechanical properties of conventional heat-cured denture base acrylic resin by incorporating different types of fillers [35]. Alhareb et al. [36] showed a 16% increase in flexural strength value compared to control samples when PMMA was reinforced with Al_2_O_3_ and ZrO_2_ with a filler concentration of 5 wt%. Moreover, the flexural modulus increased with an increase in Al_2_O_3_/ZrO_2_ nanoparticle concentration [36]. The greater value of the modulus indicates a stiffer material [16], and this improvement can be explained by a homogenous distribution of the fillers within the polymer matrix. Vojdani et al. [3] evaluated the effect of adding Al_2_O_3_ particles to PMMA denture bases on flexural strength. They found that a 6% increase in flexural strength value with 2.5 wt% Al_2_O_3_ compared to a control group could be obtained. Zhang et al. [22] investigated the effect of hybrid ZrO_2_ nanoparticles and micro-particles of aluminium borate whiskers (ABWS) at concentrations of 1 wt%, 2 wt%, 3 wt%, and 4 wt% on the flexural strength of PMMA denture base resin. They found that 2 wt% nano-ZrO_2_ with a ZrO_2_/ABWS ratio of 1:2 improved flexural strength by 32% when compared to a control group. These previous studies in the literature were in agreement with the results obtained in this study, which revealed that zirconia positively influenced the flexural properties of HI PMMA with an optimum zirconia concentration between 3 wt% and 5 wt%.

Fracture toughness (K_IC_) is a critical stress intensity factor that provides information on crack formation [29] and the ability of a material to resist crack propagation [37]. The reduction of fracture toughness in the PMMA/ZrO_2_ nanocomposites with increasing filler content could be due to a number of reasons, such as particle distribution in the polymer matrix, the type and size of the particles, the concentration of the added particles, and chemical reactions between the particles and polymer [15,35,38]. A high filler concentration leads to more filler-to-filler interactions than filler-to-matrix interactions; therefore, agglomeration may act as a point of stress concentration that could lead to non-uniform stress distribution. When applying the load, the agglomeration restrains the movement of molecular deformation and reduces the fracture toughness [38]. Sodagar et al. [39] determined that the incorporation of TiO_2_ nanoparticles to the PMMA matrix causes agglomeration, which acts as a stress raiser in the centre of the matrix and reduces the mechanical properties of the polymer material with increasing concentrations of the TiO_2_ nanoparticles. Fangqiang et al. [40] investigated the distribution of ZrO_2_ particles in PMMA matrix using two strategies during mixing: Physical method and chemical method. The physical method was conducted by melt blending, high-energy ball milling or ultrasonic vibration. In the chemical method, when mixing nanoparticles with an MMA monomer, the inorganic ZrO_2_ nanoparticles acted as a core, and the monomer as a shell structure by in situ polymerization of the monomers, known as grafting. The chemically modified nanoparticle surfaces with MMA enhanced the dispersion stability of the nanoparticles in the polymer matrix. Owing to a combined physical and chemical preparation, it was observed that the dispersion of ZrO_2_ nanoparticles in the polymer matrix was enhanced and particle aggregation and phase separation decreased to a demonstrable extent. The results of the present study on fracture toughness are consistent with those reported in the study of Alhareb et al. [36], where a PMMA denture base reinforced with 5 wt% fillers (80/20 Al_2_O_3_/ZrO_2_) showed an improvement in fracture toughness but an increase in zirconia concentration decreased toughness.

The incorporation of hard ZrO_2_ ceramic into PMMA can increase brittleness in the specimens, which would reduce the impact strength. Additionally, the lack of adhesion due to poor chemical reaction at the interface between the particles and PMMA or the inhomogeneous distribution of the nanoparticles with frequent clustering could affect the impact strength negatively [25,36]. A study conducted by Gad et al. [25] evaluated the effect of the incorporation of ZrO_2_ nanoparticles with varying concentrations (2.5 wt%, 5 wt% and 7 wt%) to PMMA denture bases on impact strength. The results showed that the impact strength decreased with an increase in ZrO_2_ nanoparticle concentration. The finding of the impact strength in the present study is in agreement with that of the previous study, with the exception of the 5 wt% ZrO_2_/PMMA nanocomposite results. This result can be explained by the fact that a concentration of 5 wt% might be the optimum quantity to improve particle distribution and reduce amalgamation. Asar et al. [35] investigated the influence of metal oxides, ZrO_2_, TiO_3_, and Al_2_O_3,_ with 1% and 2% by volume on the impact strength of the PMMA acrylic resin. In contrast to the current study, the findings showed a slight increase in the values of impact strength with 2% ZrO_2_ addition.

Denture base materials should also have adequate abrasion resistance to prevent high wear of the material by abrasive denture cleansers, food or general functional forces [41]. Greater hardness in the denture base will reduce abrasive wear. The improvement of hardness in the nanocomposites might be related to the inclusion of hard yttria-stabilized zirconia nanoparticles with fine grains, which are known as tetragonal zirconia poly-crystals (TZP). The size of the grains is dependent on the metastable nature of the tetragonal phase and can be important for providing improved mechanical properties in the nanocomposites, and this zirconia-yttria phase increases surface hardness to resist indentation [42]. However, the increase in surface hardness with the increase of the concentration of zirconia also reduces the impact strength, as seen in Table 3. The reason for hardness decrease after water immersion was described in a previous study conducted on acrylic resin denture base materials, where residual monomers release and water absorption occurring simultaneously caused the surface to become softened [43].

The finding of the present study is in agreement with a study by Yiqing et al. [43], who evaluated the hardness of PMMA/ZrO_2_ nanocomposites with different ZrO_2_ concentrations (0.5 wt%, 1 wt%, 2 wt%, 3 wt%, 4 wt%, 5 wt%, 7 wt% and 15 wt%) using indentation and pendulum hardness tests. They found that the hardness values were increased with an increase in the ratio of ZrO_2_ to PMMA, with the highest value being 15 wt%. Zhang et al. [22] investigated the effect of zirconia nanoparticles and aluminium borate whiskers (ABW) in PMMA denture bases on the surface hardness at concentrations of 1 wt%, 2 wt%, 3 wt% and 4 wt%. The results showed an increase in surface hardness with an increase in ZrO_2_/ABW content, and the optimum hardness was achieved at 3 wt% ZrO_2_ nanoparticles. They suggested that the decrease in surface hardness with higher filler loading was caused by poor adhesion of the particles to the resin matrix and filler clustering within the matrix. In another study, the incorporation of aluminium oxide (Al_2_O_3_) with percentages of 0.5 wt%, 1 wt%, 2.5 wt% and 5 wt% to PMMA acrylic resin exhibited an improvement in Vickers hardness with an increase of Al_2_O_3_ filler concentrations [3].

The lower impact strength in the nanocomposites can be related to the presence of voids and clustering of the nanoparticles [22,36]. At high magnification, the SEM images showed voids on the fractured surface, and these voids could lead to the generation of stress concentration under loading and initiate crack propagation by crossing the HI PMMA/ZrO_2_ nanocomposite matrix. At low magnification, the fractured surfaces of the nanocomposite specimens exhibited less ductile fracture compared to the control group with a large amount of fragment crack deformation, which formed an irregular surface. Furthermore, the distribution of ZrO_2_ nanoparticles in the polymer matrix was not homogeneous with evidence of agglomerations, which could reduce the impact strength, particularly at high ZrO_2_ concentrations (10 wt%).

## 5. Conclusions

With consideration to the limitations of this study, the following conclusions can be drawn:The flexural strength of the high impact (HI) heat-cured PMMA denture base was significantly enhanced by the addition of zirconia nanoparticles with 3 wt% when compared to the pure acrylic material (control group).The flexural modulus of the high impact (HI) heat-cured PMMA denture base was significantly enhanced compared to the control group by addition of zirconia nanoparticles with 1.5 wt%, 3 wt%, 5 wt% and 10 wt%. The 7 wt% of zirconia showed a non-significant enhancement compared to the control group.The fracture toughness of the zirconia-reinforced PMMA was significantly decreased, particularly at 10 wt% ZrO_2_ concentration. The fracture toughness was slightly increased at 5 wt%, but this was not significantly different compared to the control group.For all zirconia contents, the impact strength of the nanocomposites was significantly lower than that of the control group. However, at 5 wt% and 3 wt% zirconia content, the proportion of reduction in impact strength was not significantly different from that of the control group.Surface hardness continuously increased with increase of zirconia content, in the dry condition at day 0. However, in the wet condition after seven days, and 45 days surface hardness was decreased with all groups.Addition of zirconia in PMMA between 3 wt% and 5 wt% zirconia would provide the optimum mechanical properties suitable for denture base applications.

## Figures and Tables

**Figure 1 materials-12-01344-f001:**
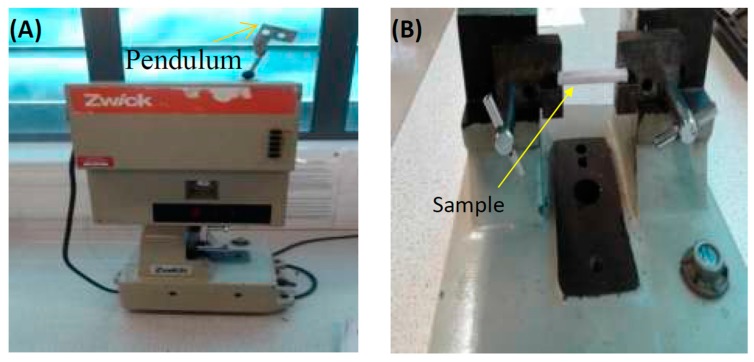
(**A**) Impact test machine and (**B**) position of sample in the machine before the test.

**Figure 2 materials-12-01344-f002:**
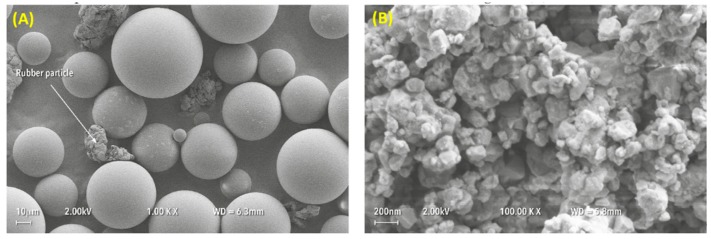
Particle size and shape distributions of (**A**) PMMA powder and (**B**) zirconia nanoparticles.

**Figure 3 materials-12-01344-f003:**
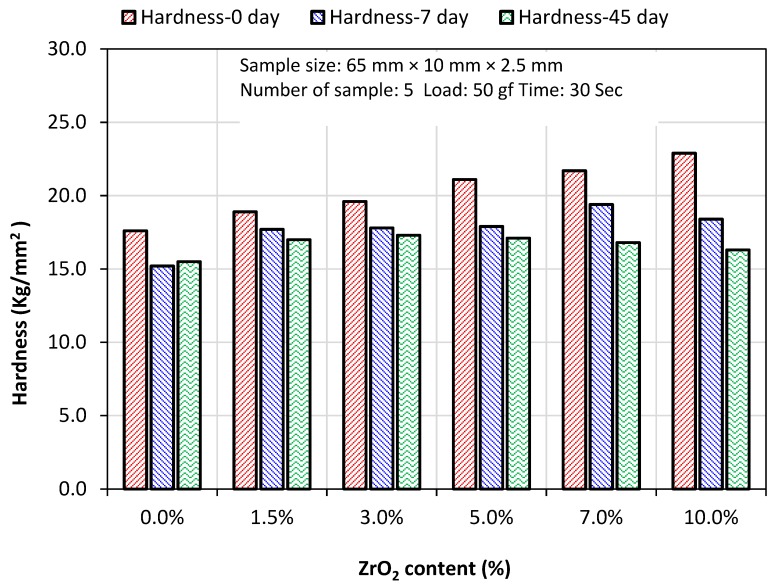
Vickers hardness median (kg/mm^2^) after 0, 7, and 45 days of water immersion.

**Figure 4 materials-12-01344-f004:**
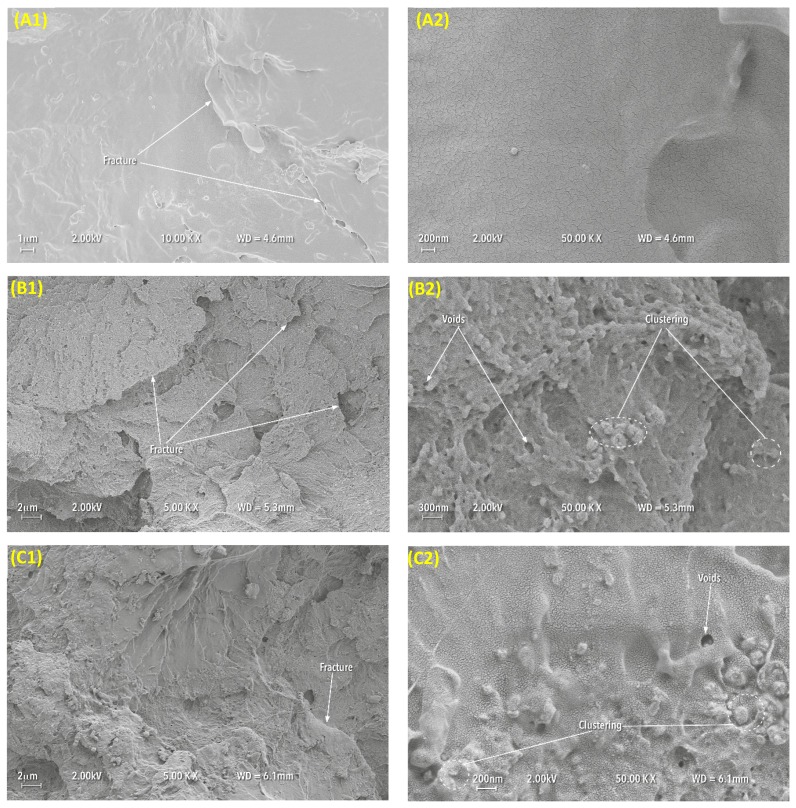
Representative SEM images of the fractured surfaces of impact strength test specimens at two different magnifications (1 at 10K and 2 at 50K for the control group (**A**) 0 wt%, (**B**) 5 wt% and (**C**) 10 wt% added zirconia, respectively).

**Table 1 materials-12-01344-t001:** Weight percent zirconia in combination with acrylic resin powder as well as monomer content of the specimen groups. HI: High impact; PMMA: Poly-methyl methacrylate; MMA: methyl methacrylate.

Experimental Groups	Zirconia (wt%)	Zirconia (g)	HI PMMA Powder (g)	HI MMA Monomer (mL)
Control	0.0	0.000	21.000	10.0
1.5	1.5	0.315	20.685	10.0
3.0	3.0	0.630	20.370	10.0
5.0	5.0	1.050	19.950	10.0
7.0	7.0	1.470	19.530	10.0
10.0	10.0	2.100	18.900	10.0

**Table 2 materials-12-01344-t002:** Mean (MPa) Standard deviation (SD) values of flexural strength, flexural modulus and fracture toughness as well as median of impact strength (kJ/m^2^) Interquartile range (IQR) for the test groups.

Zirconia Content (wt%)	Flexural Strength and SD (MPa)	Flexural Modulus and SD (MPa)	Impact Strength and (IQR) (kJ/m^2^)	Fracture Toughness and (SD) (MPa m^1/2^)
Control (0%)	72.4 (8.6) ^A^	1971 (235) ^A^	10.0 (2.69) ^A^	2.12 (0.1) ^A^
1.5	78.7 (6.9) ^A^	2237 (117) ^B^	7.03 (4.45) ^A^	1.91 (0.2) ^A^
3.0	83.5 (6.2) ^B^	2313 (161) ^B^	7.38 (4.50) ^A^	1.97 (0.2) ^A^
5.0	78.7 (7.2) ^A^	2419 (147) ^B^	9.05 (3.50) ^A^	2.14 (0.1) ^A^
7.0	72.2 (7.0) ^A^	2144 (85) ^A^	7.12 (1.50) ^A^	1.86 (0.1) ^A^
10.0	71.5 (5.7) ^A^	2204 (91) ^B^	5.89 (2.33) ^B^	1.76 (0.8) ^B^

Within a column, cells having similar (upper case) letters are not significantly different from the control (0% zirconia content) value. N = 10 specimens per group.

**Table 3 materials-12-01344-t003:** Vickers hardness (kg/mm^2^) (median and interquartile range) after 0, 7 and 45 days of water immersion.

	Day Zero (Dry)	7-Days Water- Immersion	45 Days Water-Immersion
Weight Percent Zirconia	Vickers Hardness (kg/mm^2^) Median (IQR)	Vickers Hardness (kg/mm^2^) Median (IQR)	Vickers Hardness (kg/mm^2^) Median (IQR)
Control (0.0%)	17.6 (1.7) ^Aa^	15.2 (2.0) ^Ab^	15.5 (3.3) ^Ab^*
1.5%	18.9 (3.2) ^Ab^	17.7 (1.1) ^Ab^	17.0 (1.8) ^Ab^*
3.0%	19.6 (4.0) ^Ac^	17.8 (1.2) ^Ac^	17.3 (2.8) ^Ac^
5.0%	21.1 (3.1) ^Ad^	17.9 (2.9) ^Ad^	17.1 (2.2) ^Ad^*
7.0%	21.7 (3.0) ^Be^	19.4 (0.9) ^Be^	16.8 (2.3) ^Ae^*
10.0%	22.9 (2.9) ^Bf^	18.4 (3.3) ^Bf^	16.3 (1.2) ^Af^*

Within a column, values identified using similar upper-case letters are not significantly different from the control group value; within rows values identified using the same lower-case letters are not significantly different; asterisks indicate significant differences between day 0 and 45 days; N = 5 specimens per experimental group.

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
