# Peer review of "Investigating the Mechanical Properties of ZrO2-Impregnated PMMA Nanocomposite for Denture-Based Applications"

_materials, 2019, doi:10.3390/ma12081344_

Round 1

Reviewer 1 Report

This paper investigates the mechanical properties of ZrO2 impregnated PMMA nanocomposites for denture based applications. The obtained nanocomposites were analyzed in order to have an insight into the effect of the ZrO2 filler on the mechanical properties of the polymeric matrix. The overall presentation of their efforts is good. They have described in details their methods and analyzed extensively their results. I am happy to suggest the acceptance of this work for publication after some major and minor changes.

Major comments:

1.    In the discussion, section authors discuss the state of the art. It’s advisable to discuss the same in the introduction part and authors should mention what they are doing new in this work which is different from the previously reported work.

Minor comments

2.    In table 1 specify the weight of HI MMA monomer in gm.

3.    Uniform the font size used in all the equations.

4.    The reviewer suggests modifying the SEM images for better presentation. Authors must remove the lower portion indicating the scale and magnification and manually add the scale so that it can clearly visible.

5.    Authors must indicate the direction of crack propagation in SEM images if possible.

6.    In image 3 authors must carefully edit all the images so the edited part is clearly visible.

Author Response

Hi Dear Reviewer

It was useful comments that you had gave me, and many thanks for your helping and support.

Best wishes

Saleh Zidan 

Reviewer 2 Report

Investigating the Mechanical Properties of ZrO2-Impregnated PMMA Nanocomposite for Denture based applications.

 It is a well-written paper, which is of interest for the scientific community. However, the authors should consider the following comments and suggestions:

Line 109.

The product name should be Metrocryl HI denture base powder and Metrocryl (X-linked) denture base Liquid with Metrodent being the manufacturer, please clarify.

Line 147.

Please specify the separating medium you used.

Line 197.

The description of the Impact Test would benefit from a drawing or a picture

Line 253 to 256.

The statement that the mean values of flexural modulus showed a significant increase for all specimens except 7 w% zirconia does not match with the upper case letters (A/B) in table2. May the mode of indicating significant differences in table 2 is somehow misleading.

Line 378. Just a spelling mistake: pervious

Line 420 to 430. The conclusion 1 to 3 do not correspond with the data in table 2 for the results of the statistical analysis you indicate in table 2 are different. Please clearly indicate the test series with significant differences from the control.

In their introduction the authors comprehensively described the different types of stresses that denture base materials are exposed to during function. The discussion should include a brief section that discusses the pros and cons of the zirconia induced changes in denture base material properties with respect to these different type of stresses.

Author Response

(The authors gave the same response as above.)
